# Loading a Paul Trap: Densities, Capacities, and Scaling in the Saturation Regime

**Reinhold Blümel** 

Department of Physics, Wesleyan University, Middletown, CT 06459-0155, USA; rblumel@wesleyan.edu

**Abstract:** Providing ideal conditions for the study of ion-neutral collisions, we investigate here the properties of the saturated, steady state of a three-dimensional Paul trap, loaded from a magneto-optic trap. In particular, we study three assumptions that are sometimes made under saturated, steady-state conditions: (i) The pseudopotential provides a good approximation for the number, $N_s$, of ions in the saturation regime, (ii) the maximum of $N_s$ occurs at a loading rate of approximately 1 ion per rf cycle, and (iii) the ion density is approximately constant. We find that none of these assumptions are generally valid. However, based on detailed classical molecular dynamics simulations, and as a function of loading rate and trap control parameter, we show where to find convenient dynamical regimes for ion-neutral collision experiments, or how to rescale to the pseudo-potential predictions. We also investigate the fate of the electrons generated during the loading process and present a new heating mechanism, insertion heating, that in some regimes of trapping and loading may rival and even exceed the rf-heating power of the trap.

**Keywords:** hybrid ion-neutral traps; Paul trap; ion-neutral collisions; radio-frequency heating; saturation capacity; loading rate; nonlinear; molecular-dynamics simulations

## 1. Introduction

Since its invention, about two decades ago [1,2], the ion-neutral hybrid trap has developed into a remarkably versatile tool, and excellent reviews are written about it (see, e.g., [3,4]). Consisting of a nested assembly of a magneto-optic trap (MOT) for neutrals and a radio-frequency (rf) trap for the charged-ion species, hybrid ion-neutral traps have been used successfully for the experimental investigation of cold chemistry and ion-neutral collisions by many national and international collaborations [3–15]. In the case of ion-neutral collisions, loading the rf ion trap to a saturated steady state [5,13,14] provides particularly favorable experimental conditions and is thus the focus of attention here. Previously, interesting non-monotonic dynamical regimes [13], including chaos and fractals [14], have been found to play important roles in the loading dynamics. Looking more closely at the saturation regime of a three-dimensional Paul trap (3DPT), we uncover novel facts, scaling laws, and phenomena that may result in further improvements of measured collision rates. We focus on the 3DPT, and not on the more ubiquitous linear Paul trap (LPT) [1–15] since the LPT has more parameters than the 3DPT and we are after exploring dynamical regimes and new mechanisms and phenomena that are best studied and explored in a well-defined trap, such as the 3DPT, without the need to explore a vast multi-dimensional parameter space such as in the LPT. Moreover, the results obtained are expected to carry over qualitatively to all other types of rf traps under conditions of steady-state ion loading. This has already been partially verified in [13,14], where it was shown both experimentally and theoretically with detailed simulations that, for example, the four predicted dynamical loading regimes [13] can be found in the 3DPT as well as in the LPT. To cut down even further on the number of free parameters, we focus on the (approximately) spherical 3DPT, characterized by $a = q^2/2$, where $a$ and $q$ are the two trap control parameters (see Section 2.1), which results in (approximately) equal pseudo-oscillator frequencies in all three dimensions. In this case the 3DPT is a one-parameter

system, and combined with the other free parameter, the loading rate, we are studying the loading process in a two-dimensional parameter space.

In ion-neutral collision experiments there are three assumptions that are sometimes made (see, e.g., [13]) and that we formulate here as the following three hypotheses to be tested with detailed molecular-dynamics simulations (see Section 3) of the ion dynamics in a 3DPT.

The first hypothesis to be tested is the assumption that in the saturation regime the pseudo-potential approximation provides a good approximation for the number of ions present in the trap. In Section 2.4 we show that this is not the case. We find that the actual saturation number of ions in the trap is (i) substantially smaller than suggested in the continuous-charge pseudo-potential approximation and (ii) depends on the trap control parameter $q$. In particular, the continuous-charge pseudo-potential values are not even reached in the slow-loading regime.

The second hypothesis to be tested is that the saturation regime, termed "region II" in [13,14], occurs at loading rates of about one ion per rf cycle [13,14]. In Section 2.4 we show that this is only true for trap control parameters investigated in [13,14], but is not valid in general. Instead, we show that the maximum of the saturation regime II reacts extremely sensitively to the trap control parameter $q$, ranging over several orders of magnitude in $q$.

The third hypothesis to be tested is that the density of the trapped ions in the saturation regime is approximately constant [13]. Again, we find this assumption to be problematic and show, in Section 2.5, that in most loading regimes the ion density, even in the saturation regime II, may be vastly different from a constant. The loading process accounts for density variations in the vicinity of the center of the trap, while the absorption process at the spatial limits of the trap accounts for deviations close to the maximal trapping radius of the trap. We find that under conditions of steady-state loading, a constant trapped-ion density is not even encountered in pseudo-potential approximation, that is, in a situation where there are no additional complications due to the rf heating process.

In addition to testing the three hypotheses listed above, we identify a new heating mechanism, *insertion heating* (see Section 2.3) , that, in some dynamical regimes, rivals, and may even exceed, the rf heating power in importance.

The paper is structured in the following way. The main results are presented in Section 2. Our molecular-dynamics simulation methods are defined and explained in Section 3. We conclude the paper in Section 4.

## 2. Results

In this section we present the core results of this investigation. It is divided into five subsections. In Section 2.1, to set the stage, we introduce the notation and present the equations of motion of the ions in the trap. In Section 2.2 we ask and answer the question of what happens with the electrons that are generated in the loading process. In Section 2.3 we identify and analyze the new insertion-heating mechanism. In Section 2.4, we present the results of our molecular-dynamics simulations that show significant differences with respect to pseudo-potential expectations and variations with the trap control parameter $q$. In Section 2.5 we present some typical densities as they exist in the saturation regime, illustrating significant deviations from the expected constant according to the pseudo-potential prediction.

### 2.1. Equations of Motion and Pseudopotential

If the total number of ions in the trap is $N$, then, in SI units, the equation of motion of ion number $i$ in an ideal 3DPT in the presence of the other $N-1$ ions is [16]

$$m\ddot{\vec{x}}_i(t) = -\Gamma\dot{\vec{x}}_i - 2e\left[\frac{U_0 + V_0\sin(\Omega t)}{r_0^2 + 2z_0^2}\right]\begin{pmatrix} x \\ y \\ -2z \end{pmatrix} + \left(\frac{e^2}{4\pi\epsilon_0}\right)\sum_{j\neq i}\frac{\vec{x}_i - \vec{x}_j}{|\vec{x}_i - \vec{x}_j|^3}, \tag{1}$$

where $\Gamma$ is the damping constant, $U_0$ and $V_0$ are the dc and ac voltages applied to the trap, respectively, $r_0$ and $z_0$ are the radius of the ring-electrode and the distance from the trap's center to the end-cap electrodes, respectively, and $\epsilon_0$ is the permittivity of the vacuum. Damping does not have any visible effect on the shapes of the loading curves to be discussed in Section 2.4 nor does it alter the shapes of the ion densities discussed in Section 2.5 (see also [13]). Therefore, we do not consider damping and, in what follows, set $\Gamma = 0$. Collisions with background gas are also not included in our simulations since experimentally it is always possible to reduce the rate of these collisions to negligible levels.

Introducing the dimensionless trap control parameters [16]

$$a = \frac{8eU_0}{m\Omega^2(r_0^2 + 2z_0^2)}, \quad q = \frac{4eV_0}{m\Omega^2(r_0^2 + 2z_0^2)},$$
(2)

the unit of length [17]

$$l_0 = \left[\frac{e^2}{\pi\epsilon_0 m\Omega^2}\right]^{1/3},$$
(3)

and the unit of time

$$t_0 = 2/\Omega,$$
(4)

turns (1) (with $\Gamma = 0$) into the set of dimensionless equations

$$\frac{d^2}{d\hat{t}^2}\hat{\vec{x}}_i(\hat{t}) + [a + 2q\sin(2\hat{t})]\begin{pmatrix}\hat{x}\\\hat{y}\\-2\hat{z}\end{pmatrix} = \sum_{j\neq i}\frac{\hat{\vec{x}}_i - \hat{\vec{x}}_j}{|\hat{\vec{x}}_i - \hat{\vec{x}}_j|^3},$$
(5)

which, together with the loading process, form the basis of our molecular-dynamics simulations (see Section 3). The pseudopotential is obtained by averaging over the fast oscillations of the rf trap field. It is given approximately, but explicitly, by [16]

$$U_{pp}(\hat{\vec{x}}) = \frac{1}{2}\mu_x^2(\hat{x}^2 + \hat{y}^2) + \frac{1}{2}\mu_z^2\hat{z}^2,$$
(6)

where $\mu_x$ and $\mu_z$ are the dimensionless pseudo-oscillator frequencies

$$\mu_x = \left(a + \frac{1}{2}q^2\right)^{1/2},$$
(7)

$$\mu_z = \left[2(q^2 - a)\right]^{1/2}.$$
(8)

In SI units, we have [16]

$$\omega_x = \mu_x\frac{\Omega}{2},$$
(9)

$$\omega_z = \mu_z\frac{\Omega}{2}.$$
(10)

We are exclusively focusing on the case of a spherically symmetric pseudopotential, obtained for

$$a = \frac{q^2}{2},$$
(11)

which results in

$$\mu_x = \mu_z = q.$$
(12)

We now assume that the trap is filled homogeneously with charge of charge-density $\rho^{pp}$. In pseudo-potential approximation we compute $\rho^{pp}$ in the following way. Denoting

by $Q_{\text{encl}}$ the enclosed charge, an ion a distance $r$ away from the trap's center experiences a repulsive Coulomb force

$$F_{\text{Coul}} = \frac{eQ_{\text{encl}}}{4\pi\epsilon_0 r^2} = \left(\frac{e}{4\pi\epsilon_0 r^2}\right)\left(\frac{4\pi}{3}r^3\right)\rho^{pp} = \frac{e\rho^{pp}r}{3\epsilon_0}, \tag{13}$$

which needs to be counterbalanced by the trap force

$$F_{\text{trap}} = m\omega_x^2 r = m\left(q\frac{\Omega}{2}\right)^2 r = \frac{1}{4}m\Omega^2 q^2 r. \tag{14}$$

Thus, from $F_{\text{Coul}} = F_{\text{trap}}$, we obtain, in SI units,

$$\rho^{pp} = \frac{3\epsilon_0 m\Omega^2 q^2}{4e}. \tag{15}$$

In dimensionless units, we define $\hat{\rho}^{pp}$ as the number of ions per unit volume, $l_0^3$, that is,

$$\hat{\rho}^{pp} = \frac{\rho^{pp}l_0^3}{e} = \frac{3q^2}{4\pi}. \tag{16}$$

Since any physical trap is limited in size by electrodes, other absorbing structures or dynamical processes such as the onset of chaos [14], or higher-order rf multipoles [18], we define the effective trap radius $R_{\text{cut}}$ in SI units, beyond which ions are considered absorbed, that is, lost from the trap. Defining the dimensionless trap radius,

$$\hat{R}_{cut} = R_{cut}/l_0, \tag{17}$$

the expected number of ions in the trap in the saturated regime is

$$N_s^{pp} = \frac{4\pi}{3}\hat{R}_{\text{cut}}^3 \hat{\rho}^{pp} = q^2 \hat{R}_{\text{cut}}^3. \tag{18}$$

It is convenient to relate the actual number, $N_s$, of ions in the saturated trap to the respective number of ions in the corresponding saturated pseudopotential. For this purpose we define the ratio

$$\nu_s = \frac{N_s}{N_s^{pp}} = \frac{N_s}{q^2 \hat{R}_{\text{cut}}^3}. \tag{19}$$

*2.2. Trapped Electrons?*

The ionization process in the loading region at $r \sim 0$ generates electrons at near zero kinetic energy inside of the trap. What is their fate? Are they going to be trapped or expelled from the trap?

An electron generated inside of the trap is under the influence of the trap field and the field produced by the space charge of the trapped ions. Assuming a completely filled psudopotenital (see Section 2.1), the magnitude of the confining force acting on an electron due to space charge is the same as $F_{\text{Coul}}$ in (13), that is,

$$F_{\text{conf}} = \frac{e\rho^{pp}r}{3\epsilon_0}. \tag{20}$$

For electrons, trap fields that confine ions may be considered as very slowly varying. This is also clear from the fact that the value of the trap control-parameter $q$ for electrons, $q_e$, is related to the value of the ionic trap control-parameter, $q$, according to

$$q_e = \frac{mq}{m_e} \gg 2000 \times q, \tag{21}$$

which corresponds to a strongly de-confining trap. Therefore, we may consider the trap fields acting on the electrons as quasi static, and consider only the trap field with maximum amplitude as being relevant for deciding the question of whether electrons are confined or de-confined in the combined fields of trap and ionic space charge. According to (1) the maximum trap force acting on an electron in $x$ direction is

$$F_{\text{max}} = 2e\frac{U_0 + V_0}{r_0^2 + 2z_0^2}x = \left(\frac{1}{4}a + \frac{1}{2}q\right)m\Omega^2 x. \tag{22}$$

The maximal force occurs if $x = r$. In this case, with $a = q^2/2$, we have

$$F_{\text{max}} = \left(\frac{1}{8}q^2 + \frac{1}{2}q\right)m\Omega^2 r. \tag{23}$$

The ratio of the confining force, $F_{\text{conf}}$, and the de-confining force, $F_{\text{max}}$, is

$$\frac{F_{\text{conf}}}{F_{\text{max}}} = \frac{q}{2 + \frac{1}{2}q} = \beta < 1, \tag{24}$$

since for stable 3DPT operation we need $q < 1$ [16]. This shows that under the assumption of a filled pseudo-potential, the confining force due to the ionic space charge is always smaller than the maximum of the de-confining force. This implies that as soon as they are created, the electrons are swept out of the trap within at most half of a cycle of the applied rf trap field.

In our derivation of the inequality (24) we considered only the force in $x$ direction. However, since the force in $y$ direction is the same as the force in $x$ direction, and since, according to (1), the force in $z$ direction in a 3DPT is twice as strong as the force in $x$ direction, this does not change the result that the electrons are lost from the trap essentially as fast as they are created. Thus, in both, actual trap-loading experiments and in our numerical simulations, it is safe to completely neglect the influence of the electrons generated in the photoionization process. However, the inequality (24) is not very strong in the sense that an ion plasma with a density exceeding the pseudopotential equilibrium density by a factor larger than $2/\beta = (4/q) + 1$, the factor 2 accounting for the twice stronger de-confining force in $z$ direction, is capable of trapping electrons. As shown in Section 2.5, such densities may indeed be found in the fast-loading regime in the spatial loading region close to the center of the trap, in particular for large $q$. Investigating this possibility further is beyond the scope of this paper, but is a good topic for future research. It is also possible that the electrons, on their way out of the trap, have sufficient energy to cause secondary ionizations by electron-neutral collisions. These events may be exceedingly rare, but may still deserve some further attention, if only to rule out this ionization process.

### 2.3. Insertion Heating

Whenever a new ion is created in the loading region of the trap, it is inserted into an already existing charge distribution formed by the trapped ions. In pseudo-potential approximation we may assume that the ion charge density is constant [see (15)]. Thus, inserting an additional ion into this space-charge cloud a distance $r$ away from the trap's center costs the insertion energy [19]

$$V(r) = \frac{e\rho^{pp}}{6\epsilon_0}(3R_{\text{cut}}^2 - r^2), \quad r \leq R_{\text{cut}}. \tag{25}$$

In the loading zone, that is, $r \ll R_{\text{cut}}$, we have

$$V(r) \approx \frac{e\rho^{pp}}{2\epsilon_0}R_{\text{cut}}^2, \tag{26}$$

which means that each ion insertion adds an amount of energy

$$\Delta E = \frac{e\rho^{pp}}{2\epsilon_0}R_{\text{cut}}^2 \tag{27}$$

to the trapped ion cloud. Defining $\lambda$ as the number of ions loaded per rf cycle, the insertion heating rate, $\eta$, is given by

$$\eta = \frac{\lambda\Delta E}{\left(\frac{2\pi}{\Omega}\right)} = \left(\frac{3}{4\pi}\right)E_0 q^2 \Omega\lambda\hat{R}_{\text{cut}}^2, \tag{28}$$

where we introduced the unit of energy

$$E_0 = \frac{ml_0^2}{t_0^2} = \frac{ml_0^2\Omega^2}{4} \tag{29}$$

in order to connect with the unit of energy used in [17]. The rf heating rate $H$ in [17] is dimensionless, expressed in units of $E_0/t_0$. In order to compare our insertion heating rate with the dimensionless heating rate $H$ in [17], we define the dimensionless insertion heating rate

$$\hat{\eta} = \left(\frac{t_0}{E_0}\right)\eta = \frac{2\eta}{E_0\Omega} = \left(\frac{3}{2\pi}\right)q^2\lambda\hat{R}_{\text{cut}}^2. \tag{30}$$

In [17], $H$ is graphed for various ion numbers in the trap. Therefore, it is convenient to express $\hat{\eta}$ with the help of the pseudo-potential saturation number $N_s^{pp}$ [defined in (18)]:

$$\hat{\eta} = \left(\frac{3\lambda}{2\pi}\right)\left(qN_s^{pp}\right)^{2/3}. \tag{31}$$

For the case $q = 0.2$, shown in Figure 1 of [17], this can be simplified further to

$$\hat{\eta} \approx 0.163\,\lambda\left(N_s^{pp}\right)^{2/3}. \tag{32}$$

At $q = 0.2$, the maximal rf heating rate $H^{\text{max}}$ in [17] is well approximated by

$$H^{\text{max}} = 0.20 \times \left(\frac{N_s^{pp}}{50}\right)^{5/3}, \tag{33}$$

resulting in the ratio

$$\frac{\hat{\eta}}{H^{\text{max}}} = \frac{553\lambda}{N_s^{pp}} \tag{34}$$

of the two heating rates. Assuming $\lambda = 1$, a loading rate shown in [13] to be located in the plateau region II of the loading curve for $q = 0.2$, we obtain $\hat{\eta}/H^{\text{max}} = 11.1, 5.5, 2.8$, and $1.1$ for $N_s^{pp} = 50, 100, 200$, and $500$, respectively. We see that in these cases the insertion heating rates are systematically larger than their corresponding rf heating counterparts, more than an order of magnitude on the low ion-number side. We also see that the insertion-heating advantage is reduced with increasing ion number $N_s^{pp}$. Nevertheless, for the cases studied here, insertion heating is certainly the dominant heating mechanism. For $N_s = 50$, and reducing the loading rate one order of magnitude to $\lambda \sim 0.1$, the two heating rates are of the same order of magnitude, and rf heating dominates for $\lambda \sim 0.01$. This range of loading rates is, in fact, part of the range of loading rates studied in detail in the following Section 2.4.

## 2.4. Saturation Curves

So far, we studied the number $N_s$ of ions in the saturation regime in the context of the static pseudo-potential approximation. We now switch on the rf term in (1) and

study the steady-state ion capacity, $N_s(\lambda)$, under saturation conditions for the combined time-dependent processes of rf drive and ion loading at a loading rate $\lambda$, where $\lambda$ is the number of ions loaded per rf cycle.

Actual hybrid traps contain of the order of $10^5, \ldots, 10^6$ ions [14]. While it may be possible to simulate the loading and saturation dynamics of such a trap for a single choice of trap control parameters and loading rates, detailed simulations for ranges of parameters, as, for example, presented in this subsection, are impractical to do for more than $\sim 10^3$ stored ions with the available computer equipment. Therefore, we need to keep the number of particles down so that extensive simulations can be performed in a reasonable time. A good compromise was to scale down the actual experimental trap dimensions by about a factor 10 and define our model trap with a loading-zone radius of $\hat{R}_L = 3$ and a trap size that we vary in steps of 5 from $\hat{R}_{\text{cut}} = 15$ to $\hat{R}_{\text{cut}} = 30$. In Figure 1 we show examples of loading curves, similar to the ones obtained in [13,14], for $\hat{R}_{\text{cut}} = 15$ and loading rates ranging from $\lambda = 0.01$ to $\lambda = 100$. In contrast to the loading curves in [13,14], which show ion saturation numbers $N_s(\lambda)$ as a function of loading rate $\lambda$, Figure 1 shows scaled ion numbers

$$\nu_s(\lambda) = \frac{N_s(\lambda)}{N_s^{pp}}, \tag{35}$$

where $N_s^{pp}$, the ion number expected in the pseudo-potential approximation, is defined in (18) and $N_s(\lambda)$ is the actual number of ions in the trap according to our simulations. Scaling is convenient since this way we are able to show loading curves for a wide range of trap parameters ($q = 0.1, \ldots, 0.4$) in the same frame. The scaling also provides us with the advantage of a direct comparison of the actual ion saturation number with the pseudo-potential prediction.

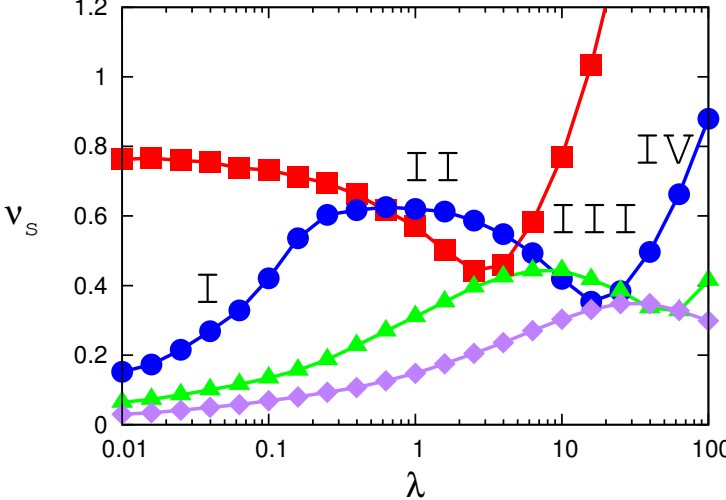

**Figure 1.** Scaled saturation ion number $\nu_s(\lambda) = N_s(\lambda)/(q^2 \hat{R}_{\text{cut}}^3)$ for four different values of the trap control parameter $q$ as a function of loading rate $\lambda$. Red squares: $q = 0.1$; blue circles: $q = 0.2$; green triangles: $q = 0.3$; purple diamonds: $q = 0.4$. All four loading curves show at least three of the four dynamical regimes predicted in [13,14] and explicitly marked I, II, III, and IV in the case of the curve corresponding to $q = 0.2$ (blue circles). For increasing $q$ we observe that the height of the plateaus decreases and the maxima of the plateaus shift over several orders of magnitude toward larger loading rates.

Figure 1 confirms the universal shape of loading curves predicted in [13,14] to occur for all rf traps. Focusing on the loading curve for $q = 0.2$, the four dynamical regimes are marked in Figure 1 with the Roman numerals I, II, III, IV, respectively. Dynamical regime I corresponds to the slow-loading regime. Ions are loaded from the MOT into the 3DPT with near-zero initial kinetic energy, and are, therefore, initially, trapped deeply in

the potential well of the trap. Subsequently, they slowly gain energy due to the combined action of rf- and insertion heating. Having gained enough energy from these two heating mechanisms, they escape from the trap. The saturation ion number in this regime is determined by the balance between the rate of newly loaded ions and the rate of escape dictated by the heating rates. Increasing the loading rate, more ions can simultaneously undergo this process of loading, heating, and escaping, which explains why the region-I loading curve rises monotonically with increasing loading rate. At some point, however, the increase of the number of ions in the trap with increasing loading rate has to stop, and we enter dynamical region II. In this region, the role of heating is diminished, and the role of Coulomb-Coulomb interactions is emphasized, roughly resulting in a situation of a strongly coupled (Coulomb) fluid (plasma), where each new "drop" (ion loaded) overflows the "bucket" (the trap) such that the total number of fluid (ions) contained in the bucket (trap) cannot exceed the maximum capacity of fluid in the bucket (ions in the trap). This situation results in the quasi-stationary behavior (plateau) that characterizes dynamical region II (see Figure 1). Increasing the loading rate even further corresponds to a situation where fluid (ions) accumulate with a density larger than the average trap density in the loading region (MOT region) of the trap, actively pushing ions out of the trap due to strong Coulomb forces. This results in the dip of the loading curves, marked as dynamical region III in Figure 1. Further increase of the loading rate piles up so much inertially confined charge in the loading region that a highly charged "core" develops in the trap (see Section 2.5), that accelerates all charge outside of the core swiftly and ballistically out of the trap. This explains the sharp rise of the loading curve in dynamical region IV (see Figure 1). Dynamical region IV is currently the best understood region and resulted in an experimentally confirmed behavior of $\nu_s \sim \lambda^{2/3}$ [13].

In Figure 1 all four dynamical regimes are present for $q = 0.2$ and $q = 0.3$. However, in Figure 1 our $\lambda$ scale does not extend far enough to the left and to the right to see all four dynamical regimes for $q = 0.1$ and $q = 0.4$. But we still see regions II, III, and IV for $q = 0.1$, and regions I, II, and the beginnings of region III for $q = 0.4$. This confirms expectations. More importantly, however, we see that only in the fast-loading regime, not currently used for ion-neutral collision experiments, do the loading curves reach the level of 1 (and beyond) predicted by the pseudo-potential approximation. In all other dynamical regimes, $\nu_s$ is significantly smaller than 1, indicating that the static pseudo-potential approximation overestimates the number of ions stored in the trap. So, we have a first result: The saturation numbers do not agree with the pseudo-potential prediction in two important ways: (i) Instead of being constant, $\nu_s(\lambda)$ shows a non-monotonic structure, already found in [13,14]. (ii) Even at the maximum, $\nu_s(\lambda)$ is significantly below the pseudo-potential prediction. We also notice that for increasing $q$, the maxima of the loading curves in dynamical regime II shift very quickly toward larger loading rates.

The depression of the saturation ion levels with respect to the pseudo-potential prediction is investigated further in Figure 2. This figure shows the scaled maxima of the loading curves of Figure 1,

$$\nu_s^{\mathrm{max}} = \frac{N_s^{\mathrm{max}}}{q^2 \hat{R}_{\mathrm{cut}}^2}, \tag{36}$$

for several values of $q$ and $\hat{R}_{\mathrm{cut}}$. Figure 2 confirms that the scaled saturation levels are substantially below 1 and reach down to $\approx 0.4$, that is, a trap capacity of only about 40% of the pseudo-potential prediction, at $q = 0.4$. Even for small $q$, the trap capacity is only about 80% of the pseudo-potential prediction. However, we also see that the ion capacity scales $\sim \hat{R}_{\mathrm{cut}}^3$. Therefore, the scaled $\nu_s^{\mathrm{max}}$ is approximately independent of $\hat{R}_{\mathrm{cut}}$ and depends only on $q$. A scaling function may be derived from Figure 2, which can be used to calibrate $N_s(\lambda)$ with respect to the pseudo-potential prediction.

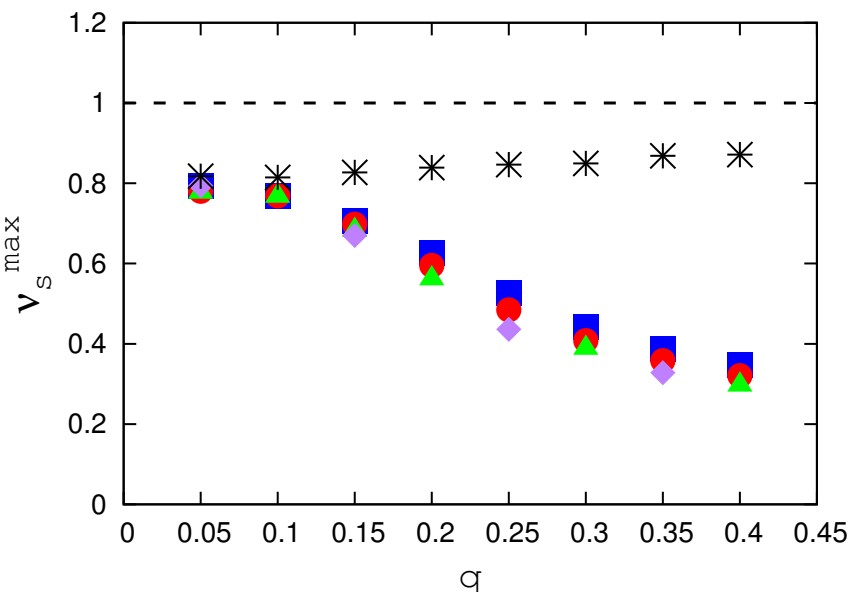

**Figure 2.** Normalized scaled maxima $\nu_s^{\max}(q) = N_s^{\max}/(q^2\hat{R}_{\text{cut}}^3)$ of the saturated ion numbers as a function of trap control parameter $q$. The black dashed line indicates the static, filled-pseudo-potential expectation of the normalized particle number at saturation. The black stars are the $\nu_s^{\max}(q)$ values obtained in the case where the rf-driven trap is replaced with its corresponding static pseudo-potential. The filled plot symbols are the $\nu_s^{\max}(q)$ values resulting from full three-dimensional molecular-dynamics simulations of the trapped ions under the condition of constant-rate loading. Blue squares, red circles, green triangles, and purple diamonds correspond to $\hat{R}_{\text{cut}} = 15, 20, 25, 30$, respectively. The approximate overlap of the plot symbols indicates near-perfect $1/\hat{R}_{\text{cut}}^3$ scaling.

The rapid shift of the saturation plateaus to the right as $q$ increases is illustrated in Figure 3, which shows the loading rate $\lambda^{\max}$ at which the loading-plateau maxima occur (see Figure 1) for several different $q$ values. In the region from 0.1 to 0.3 we see an exponential dependence of $\lambda^{\max}$ on $q$, approximately given by (see straight, blue line in Figure 3):

$$\lambda^{\max} \approx 0.01 \times 10^{16(q-0.1)}. \tag{37}$$

In [13] we found that for $q = 0.2$ the maximum of $N_s(\lambda)$ occurs at $\lambda \sim 1$. Figure 3 shows that this is not always so. Quite the contrary: Indicated by closely following the straight blue line in Figure 3, representing the function (37), $\lambda^{\max}$ shows extreme (exponential) sensitivity to the trap control parameter $q$ in the region $0.1 < q < 0.3$. The deviation from exponential behavior in the region $q > 0.3$ may be due to the fact that for increasing $q$ the maximum in region II turns into a shoulder on the region-IV curve (see Figure 1), masking a possible continuation of the exponential behavior. However, a clearer understanding of the deviation from exponential behavior has to await further analytical progress. The deviation of the data point at $q = 0.05$ has a simpler explanation. It is due to the fact that our simulations are cut at $\lambda = 0.01$ and thus do not extend far enough to the left to capture the true location of the maximum of the loading curve at $q = 0.05$.

In this subsection we compared actual steady-state saturation ion numbers obtained from molecular-dynamics simulations with the ion numbers expected from a model that assumes that the pseudo-potential is filled with a homogeneously distributed charge. In reality, however, the ions in the trap are localized, discrete charges, which, in analogy to spherical Coulomb crystals [20] leads to a reduced ion-number capacity in the trap. Therefore, it is possible that the agreement between the pseudo-potential approximation and the simulations may be improved somewhat by taking the discretization of charge into account. However, even in this improved pseudo-potential model, the decrease of $\nu_s^{\max}$ with increasing $q$, observed in Figure 2, cannot be accounted for.

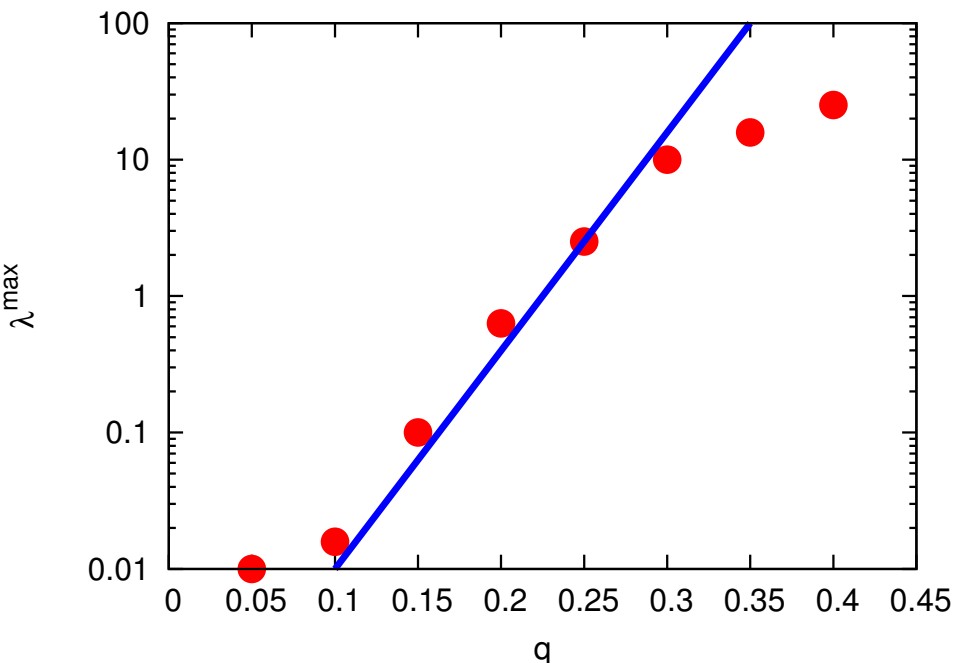

**Figure 3.** $\lambda^{\max}$ as a function of $q$, illustrating how the loading rate, $\lambda^{\max}$, at which the number of the particles in the trap assumes its maximum, shifts toward higher loading rates as a function of increased $q$. Indicated by the straight blue line, representing the function $\lambda^{\max} = 0.01 \times 10^{16(q-0.1)}$, approximately exponential sensitivity is observed in the range $0.1 < q < 0.3$.

### 2.5. Densities

Because of its quadratic nature, the static pseudopotential discussed in Section 2.1 predicts that the charge density $\hat{\rho}^{pp}$, given in (16), is constant throughout the trap. A constant density is important for ion-neutral collision experiments, since only this way we may assume homogeneous and isotropic conditions throughout the reaction volume.

In [13,14] it was assumed that the best conditions in terms of charge density should occur at the maxima of the loading curves (see Figure 1). To test this assumption, Figure 4 shows the charge densities $\hat{\rho}(\hat{r})$ (histograms) obtained as a result of full three-dimensional molecular-dynamics simulations (see Section 3) for $q = 0.1, 0.2, 0.3, 0.4$ as a function of the dimensionless radial position $\hat{r} = |\hat{\vec{r}}| = \sqrt{\hat{x}^2 + \hat{y}^2 + \hat{z}^2}$. Also shown, for comparison, is the constant-density pseudo-potential prediction (red, horizontal line). We see that, while, at least in some regions, the actual densities are close to the pseudo-potential prediction, the condition of constant charge density is approximately satisfied over a large range of the trap volume only for $q = 0.1$, the smallest of the $q$ values shown in Figure 4. And even in this case, we see a significant deviation from a constant density, especially toward the radius of the trap at $\hat{R}_{\text{cut}}$. In fact, the deviation from a constant density at $\hat{r} \approx \hat{R}_{\text{cut}}$ accounts for much of the missing ions (see Figure 2) as compared to the pseudo-potential prediction. We see this in the following way. In a simple model of the density shown in Figure 4a, we assume that the density is constant and equal to $\hat{\rho}^{pp}$ in $0 \le \hat{r} \le \hat{R}_1$, decreases linearly in $\hat{R}_1 \le \hat{r} \le \hat{R}_2$, and reaches 0 at $\hat{r} = \hat{R}_2$. Thus, in this model, the density is given by

$$\hat{\rho}(\hat{r}) = \hat{\rho}^{pp} \times \begin{cases} 1 & \text{for } 0 \le \hat{r} \le \hat{R}_1, \\ \frac{\hat{R}_2 - \hat{r}}{\hat{R}_2 - \hat{R}_1} & \text{for } \hat{R}_1 \le \hat{r} \le \hat{R}_2. \end{cases} \tag{38}$$

The number of particles in this model is

$$N_s = \hat{\rho}^{pp}\left(\frac{4\pi}{3}\right)\hat{R}_1^3 + 4\pi \int_{\hat{R}_1}^{\hat{R}_2} \hat{\rho}(\hat{r})\hat{r}^2 \, d\hat{r} = \left(\frac{\pi}{3}\right)(\hat{R}_1^3 + \hat{R}_1^2\hat{R}_2 + \hat{R}_1\hat{R}_2^2 + \hat{R}_2^3)\,\hat{\rho}^{pp}. \tag{39}$$

Therefore, if we now normalize the actual number of stored ions, $N_s$, to the pseudo-potential prediction, $N_s^{pp}$, we obtain

$$\nu_s = \frac{N_s}{N_s^{pp}} = \left(\frac{1}{4\hat{R}_2^3}\right)(\hat{R}_1^3 + \hat{R}_1^2\hat{R}_2 + \hat{R}_1\hat{R}_2^2 + \hat{R}_2^3). \tag{40}$$

Inserting $\hat{R}_1 = 20$ and $\hat{R}_2 = 25$, appropriate for the case of Figure 4a, we obtain $\nu_s = 0.74$, close to the actual observed ratio of $\nu_s \approx 0.8$, the small hump in the density occurring before the onset of the decline of the density at $\hat{R}_1 = 20$ accounting for much of the difference. So, we now have an explanation for the missing ion numbers: For small $q$, such as in Figure 4a, the missing ions are due to the behavior of $\hat{\rho}(\hat{r})$ close to the cut-off at $\hat{r} = \hat{R}_{\text{cut}}$.

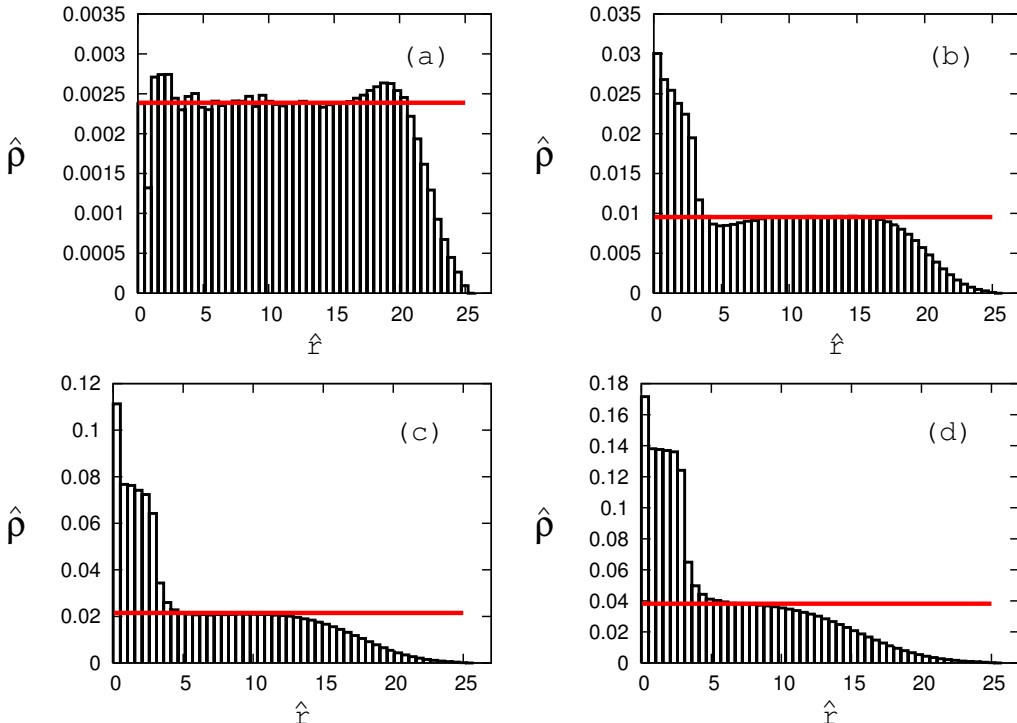

**Figure 4.** Ion densities $\hat{\rho}$ as a function of distance $\hat{r}$ from the trap's center (histograms) for four different trap parameters, that is, $q = 0.1$ (**a**), $q = 0.2$ (**b**), $q = 0.3$ (**c**), and $q = 0.4$ (**d**) at their respective loading rates $\lambda^{\text{max}}$ (see Figure 3). The red, horizontal line is the pseudo-potential prediction for $\hat{\rho}$. We see that the pseudopotential predicts the density approximately only for $q = 0.1$, while significant deviations from the pseudo-potential prediction are observed for $q > 0.1$.

We now turn to the densities corresponding to larger values of $q$. Figure 4b shows the case for $q = 0.2$. Contrary to the case of $q = 0.1$, where the loading-curve maximum corresponding to the density shown in Figure 4a occurs at a loading rate of $\lambda^{\text{max}} = 0.01$, the loading rate at the maximum of the loading curve in the case of $q = 0.2$, corresponding to the density shown in Figure 4b, now occurs at $\lambda^{\text{max}} = 2.5$. Thus, the loading rate in the case of Figure 4b is a factor 250 larger than the loading rate in Figure 4a, although the trap control parameter only doubled. This, of course, is a result of the exponential sensitivity of the loading rate in $q$ as shown in Figure 3 and quantified in (37). Accordingly, reflecting the vastly larger loading rate, we now see a pile-up of density in the loading region $0 \leq \hat{r} \leq \hat{R}_L = 3$. The pile-up of charge in the loading zone may be explained in the following way. Charge created in the region $\hat{r} \leq \hat{R}_L$ needs to be transported away from the loading zone. One mechanism to ensure this is rf heating, which imparts kinetic energy on the ions in the loading zone. However, since rf heating results from the

micro-motion, and the micro-motion amplitude of an ion is proportional to its distance $\hat{r}$ from the trap's center [16], rf heating is not very effective close to the trap's center, that is, in the loading zone, resulting in the observed pile-up of charge in the loading zone. Consequently, together with its behavior at the edges of the trap, the ion density in the trap is now significantly different from the predicted constant both in the vicinity of $\hat{r} \approx 0$ and around $\hat{r} \approx \hat{R}_{\text{cut}} = 25$. As shown in Figure 4c,d, this trend continues for larger $q$ values, corresponding to ever larger loading rates, according to Figure 3, so that the $\hat{r}$-interval of constant density shrinks with increasing $q$ and is near negligible in the case of $q = 0.4$ [see Figure 4d]. We also see that with increasing $q$ the radius $\hat{R}_1$ in our model above decreases significantly, explaining the depression of the ion-storage capacity of the trap down to about 40% of the pseudo-potential prediction at $q = 0.4$ (see Figure 2). Apparently, the overshoot of the density in the loading region $0 \leq \hat{r} \leq \hat{R}_L = 3$ cannot compensate for the ion loss due to the decline of the density toward $\hat{r} = \hat{R}_{\text{cut}}$.

What causes the decrease of the density close to $\hat{R}_{\text{cut}}$? Since the 3DPT is an rf trap, the micro-motion comes to mind as a possible direct explanation. According to [16] the micro-motion amplitude is given by

$$\Delta \hat{r}_{mm} = \frac{q\hat{r}}{2},\tag{41}$$

and is proportional to both $q$ and $\hat{r}$. Therefore, if $\Delta \hat{r}_{mm}$ would be of the order of $\hat{R}_{\text{cut}} - \hat{R}_1$, one could immediately explain the missing ions in the vicinity of $\hat{R}_{\text{cut}}$ as being knocked out of the trap directly by the micro-motion. Moreover, since $\Delta \hat{r}_{mm}$ is proportional to $q$, this would even explain why more ions are missing for increasing $q$. However, according to the following reasoning, the micro-motion amplitude can be ruled out as the direct cause for the missing ions. At $\hat{R}_{\text{cut}} = 25$ and $q = 0.1$, the micro-motion amplitude works out to be $\Delta \hat{r}_{mm} = 1.25$ and increases only to $\Delta \hat{r}_{mm} = 5$ for $q = 0.4$. So, while the micro-motion amplitude is certainly not negligible, and certainly plays a role in carrying ions across the boundary at $\hat{r} = \hat{R}_{\text{cut}}$, its magnitude is too small to explain the shape of the density in Figure 4 via this direct knock-out mechanism. However, the relatively large micro-motion amplitude in the vicinity of $\hat{R}_{\text{cut}}$ indicates that the time-dependence of the trap field may play a major role in explaining the missing ions. To nail this down, in Figure 2 we compare the ion saturation numbers for the time-dependent 3DPT with the ion saturation numbers obtained from fully three-dimensional molecular-dynamics simulations (see Section 3) where, in addition to ion loading, only the time-independent, static pseudopotential was switched on (stars in Figure 2). We see that, although even in the dynamically loaded pseudo-oscillator trap the saturation numbers are smaller than predicted by the static pseudo-potential approximation without dynamic ion loading, the densities obtained from the the pseudo-oscillator simulations for large $q$ are significantly above those for the fully time-dependent 3DPT, and merge with those of the 3DPT only in the small-$q$ regime, in which the rf heating power is small. This proves conclusively that the main reason for the missing ions is strong rf heating at large values of $\hat{r}$ close to $\hat{R}_{\text{cut}}$, and in addition explains, because of the proportionality of $\Delta \hat{r}$ to both $\hat{r}$ and $q$ why the effect of the missing ions increases both with increasing $\hat{r}$ and increasing $q$. Thus, the decrease of the density in $\hat{R}_1 \leq \hat{r} \leq \hat{R}_2$ is dynamical in origin, that is, it is caused by the combined effects of rf heating and the flow of ions generated in the loading zone and absorbed at $\hat{r} = \hat{R}_{\text{cut}}$. That the flow itself is an important mechanism is shown by the fact that inspection of the densities obtained from the simulation of loading the pseudopotential shows the same linear decrease of the density in the region between $\hat{R}_1$ and $\hat{R}_2 = \hat{R}_{\text{cut}}$, only that in the pseudo-potential case $R_1$ is significantly closer to $\hat{R}_{\text{cut}}$ than in the case of the time-dependent trap, accounting for the significantly larger ion saturation numbers in the pseudo-potential case (stars in Figure 2). Since the flow is important, a combined model consisting of heating and hydrodynamic transport components might explain the different shapes of the charge densities in the different loading regimes. This, however, is beyond the scope of this paper.

The temperature is also thought to be constant throughout the trap at saturation. Again, this may not be true, since the micro-motion amplitude, $\Delta \hat{r}_{mm}$, and therefore the effectiveness of rf heating, depends on the distance $\hat{r}$ from the trap's center. Therefore, the local temperature may be a function of distance from the center of the trap. This, too, is a promising topic for future research.

## 3. Methods

The numerical simulations are performed under the assumption that the ions in the Paul trap, each of mass $m$ and charge $+e$, are classical, Newtonian particles, following Newtonian trajectories, subject to the Paul-trap forces. Given suitable initial conditions of the ion trajectories, provided by the loading process (see below), the system of Equation (5) is at the core of our simulations. Since we are looking for qualitatively new phenomena in the loading process of the Paul trap, speed of the numerical integration of the system (5) is not as important as robustness and reliability of the results. Therefore, a simple 4th order Runge-Kutta method [21] with constant step size is chosen to integrate (5), which has proven in the past to be both sufficiently fast and numerically stable [16,17,22,23]. Higher-order and faster integration methods certainly exist [21]. However, in exploratory simulations, a robust method, such as the constant step-size 4th order Runge-Kutta method is preferred.

Dynamical loading of the trap, in particular in the saturated, steady state, is accomplished in the following way. We first define the loading zone, which for the current simulations is a ball of radius $\hat{R}_L$ at the center of the trap. At time $\hat{t}_k$ an ion is created with velocity zero inside of the ball. This is a good approximation for MOT-loaded ions [24]. It was checked previously [13] that loading with an initial velocity, for instance drawn from a thermal distribution, makes little difference in the results. With $\hat{t}_k$, we also determine the time $\hat{t}_{k+1}$ for the next ion to be created in the future. The loading times $\hat{t}_k$, $k = 1, 2, \ldots$, are assumed to be uncorrelated and drawn from a Poisson distribution such that the expectation value of $\Delta \hat{t}$, $\langle \Delta \hat{t} \rangle = \langle \hat{t}_{k+1} - \hat{t}_k \rangle = \pi / \lambda$, where $\Delta \hat{t}$ is the time interval between loading events and $\lambda$ is the loading rate, that is, the number of ions loaded per rf cycle. Concerning their spatial distribution, we assume that the particles are created at random positions with a uniform spatial distribution within the loading-ball volume. It was checked that loading with a Gaussian spatial distribution does not make any qualitative difference in the results. In the time interval between loading events, that is, from $\hat{t}_k$ to $\hat{t}_{k+1}$, the particles in the trap are governed by the equations of motion (5). Once arrived at $\hat{t}_{k+1}$, and before creating the next ion at $\hat{t} = \hat{t}_{k+1}$, the algorithm checks whether one or more particles have crossed the absorbing boundary of the trap, located at $\hat{R}_{\text{cut}}$. Following instantaneous deletion of all particles that exceed $\hat{R}_{\text{cut}}$, the next particle is loaded at $\hat{t} = \hat{t}_{k+1}$, $\hat{t}_{k+2}$ is determined, and the solution algorithm for (5) is restarted with the new, adjusted ion number. This procedure is followed for as long as it takes to reach a pre-specified simulation time that is long enough so that the ion number in the trap has settled down to its saturated, asymptotic value. Following this method, the data in Figures 1–3 was generated.

Concerning Figure 4, the space in the interval $0 \leq \hat{r} \leq \hat{R}_{\text{cut}} = 25$ was divided into 50 equi-spaced intervals $\Delta_j = [\hat{r}_{j-1}, \hat{r}_j]$, $\hat{r}_j = j/2$, $j = 1, \ldots, 50$. Once the simulation reached steady state, data was taken for $N_d$ rf cycles in the steady-state regime, where $N_d$ was a large number of the order of $10^5$. After each rf cycle in the steady-state data-taking regime, the number of ions in each of the 50 $\Delta_j$ intervals was accumulated in a corresponding histogram bin. At the end of the $N_d$ data-taking cycles, the density $\hat{\rho}$ was then computed by normalizing the data in the 50 $\Delta_j$ intervals and displayed in the way of a histogram in Figure 4.

## 4. Conclusions

In Section 2.2 we started with a topic that is often overlooked: The fate of the electrons that are generated when loading an rf trap from a MOT. Since the ions stored in the rf trap represent a strong positive space charge, and since the electrons in the photo-

ionization process are generated with essentially zero kinetic energy, the question whether the electrons are potentially trapped by the positive space charge is not so far-fetched. We showed in Section 2.2 that in the slow-loading regime we can always safely neglect the created electrons, since, despite the presence of a substantial positive space charge, the electrons are pulled out of the trap by the trap's rf field within at most half an rf cycle. However, in the fast-loading regime, the space charge built-up, especially in the loading zone close to the center of the trap, may be strong enough to result in bound electrons. We did not follow up on this promising line of research. So, the question of whether there may be loading regimes with bound electrons is still open. This topic broadly falls into the area of ultra-cold neutral plasmas (see, e.g., [25]) and tools and methods from this field may eventually be useful to decide this open question. In Section 2.3 we studied a new heating mechanism, that is, *insertion heating*. According to this mechanism, each ion generated from a neutral MOT is inserted into the already existing space charge formed by the sea of trapped ions and thus contributes Coulomb energy that represents a source of heating that adds to the well-known rf heating of a Paul trap [16,17]. We saw that, compared to rf heating, and depending on trap and loading parameters, insertion heating may be very strong and needs to be taken into account for a full understanding of the loading process. All conclusions drawn in Section 2.3 are based on approximate analytical arguments. The next step is to study this mechanism in more detail with molecular-dynamics simulations. However, before this can meaningfully be accomplished, protocols have to be developed on how to, computationally, separate insertion heating from rf heating. This is a topic for future research. In Section 2.4 we investigated saturation curves as a function of loading rate for several different $q$ values. We scaled the curves to their respective pseudo-potential predictions to better observe their nonlinear behavior and the changes as a function of $q$. In particular, we observed that, contrary to a previous suggestion [14], the region-II plateau of the saturation curves does not always occur for loading rates of about 1 ion per rf cycle, but its position as a function of loading rate shifts rapidly over several orders of magnitude in the loading rate. In Section 2.4 we presented the maximum of scaled ion numbers in region II as a function of $q$. We showed that the expected number of ions at saturation is significantly smaller than its pseudo-potential expectation. A clue of where this may come from was presented in Section 2.5, where we studied the shapes of the ion density distributions in the trap as a function of distance from the trap's center. Far from being constant, as expected on the basis of the static pseudo-potential approximation, the densities show an inhomogeneous shape, which even persists in the slow-loading regime. Therefore, corroborated by detailed, microscopic molecular-dynamics calculations, we conclude that the pseudo-potential approximation to the saturation properties of a Paul trap, while a good starting point, needs to be corrected by the use of appropriate re-normalization functions. While we focused on the loading dynamics of the 3DPT with spatially uniform loading, it was found in [13] that the ion dynamics and consequently the predicted collective phenomena, such as, for example, the existence of the four dynamical regimes (see Figure 1), are independent of the particularities of the loading process, the shape of the loading region, and the type of trap in use (both 3DPT and LPT were studied). Therefore, we are confident that the observations and conclusions drawn will carry over qualitatively to all other types of hybrid traps in use for the study of ion-neutral collisions.

**Funding:** This research received no external funding.

**Acknowledgments:** The author thanks D. S. Goodman and W. W. Smith for the opportunity to contribute to the special issue on ion-neutral collisions.

**Conflicts of Interest:** The author declares no conflict of interest.

## Abbreviations

The following abbreviations are used in this manuscript:

3DPT     three-dimensional Paul trap
LPT       linear Paul trap
rf         radio frequency
MOT     magneto-optic trap

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
