# Peer review of "Loading a Paul Trap: Densities, Capacities, and Scaling in the Saturation Regime"

_atoms, doi:10.3390/atoms9010011_

Round 1
Reviewer 1 Report
The paper deals with molecular dynamics simulations of a MOT-loaded Paul trap. In particular, it tests three assumption which are usually made by researchers in this area, and proposes a new heating mechanism for the ions. The paper is globally well written and contains interesting results. In my opinion, it could be accepted for publication in Atoms, with a couple of small amendments.
My detailed comments are:
- – The paper is written in quite an unconventional format, with two consecutive introductions, section 1 and the beginning of section 2. This second introductory text is actually a kind of Concluding section, where the main results are stated. I would prefer the use of a more conventional format, with the conclusions at the end. But given the clarity of the paper, I think that the present version could eventually be accepted.
- – The discussion of trapped electrons is not completely convincing, but their neglect is a good first approximation. Trapped electron populations could be estimated by using, for instance, the Thomas-Fermi model proposed in a recent molecular dynamics paper by R. Ayllon et al, Phys. Plasmas, vol. 26, 033501 (2019). This reference could eventually be relevant to the present research.
- – The proposed new heating mechanism is very interesting, but the absence of any numerical evidence in the present simulations is quite striking. A short comment should clarify this point.
Author Response
Detailed response to referee 1's report
=======================================
In the referee's opinion, the paper
"... could be accepted for publication in Atoms,
with a couple of small amendments."
In particular, the referee would like to see
the following three points addressed:
1. The paper is written in quite an unconventional format,
with two consecutive introductions, section 1 and the beginning of section 2.
This second introductory text is actually a kind of Concluding section,
where the main results are stated.
I would prefer the use of a more conventional format,
with the conclusions at the end. But given the clarity of the paper,
I think that the present version could eventually be accepted.
2. The discussion of trapped electrons is not completely convincing,
but their neglect is a good first approximation.
Trapped electron populations could be estimated by using,
for instance, the Thomas-Fermi model proposed in a recent
molecular dynamics paper by
R. Ayllon et al, Phys. Plasmas, vol. 26, 033501 (2019).
This reference could eventually be relevant to the present research.
3. The proposed new heating mechanism is very interesting,
but the absence of any numerical evidence in the present
simulations is quite striking.
A short comment should clarify this point.
These are three excellent points, which I now
address in full:
1. Answer:
----------
This point is well taken. The unconventional format
is a result of the article structure required by the journal,
i.e., Introduction, Results, Methods, optional Conclusions,
which is not optimal for a theory paper.
Nevertheless, I agree with the referee that making use
of the optional Conclusions section does improve the
presentation.
Action taken:
-------------
What the referee refers to as the second introduction
has been reworked into the new Conclusions section.
What used to be the second introduction has been
significantly shortened.
2. Answer:
----------
This is a valuable hint.
Action taken:
-------------
In the Conclusions section, after stating
that the trapping of electrons is still
an open question, I added the following sentence:
"This topic broadly falls into the area of
ultra-cold neutral plasmas
(see, e.g., [25]) and tools
and methods from this field may eventually
be useful to decide this open question."
Here, [25] is the added reference pointed out
by the referee.
3. Answer:
----------
Insertion heating is a new mechanism and
appropriate simulation
protocols still need to be developed.
An important question to resolve in
future research is how to computationally
separate insertion heating from rf heating.
Action taken:
-------------
I inserted the following sentences
in the Conclusions section following the
presentation of the results of subsection 2.3:
"All conclusions drawn in Sec. 2.3 are
based on approximate analytical arguments.
The next step is to study this mechanism
in more detail with molecular-dynamics simulations.
However, before this can meaningfully be accomplished,
protocols have to be developed on how to,
computationally, separate
insertion heating from rf heating. This is a topic
for future research."
Reviewer 2 Report
The manuscript discusses the properties of the saturated, steady state of a magneto-optic trap-loaded three-dimensional Paul trap based on three assumptions. The paper investigated the convenient dynamical regimes for ion-neutral collision experiments and provides a solution of how to rescale to the pseudo-potential predictions. This work also investigated the fate of the electrons generated during the loading process and present a new heating mechanism, and carried out a numerical simulation assuming the ions in the Paul trap. The analysis and the description throughout the paper is well comprehensive. The results are interesting and can be considered for publication in atoms. However, there are some comments which should be addressed:
- Abstract, line 2: the author should write a magneto-optic trap instead of a MOT.
- Four dynamical regimes are not clearly defined, especially, in Fig. 1 four dynamical regimes should be clearly marked to Increase readability.
- Symbols in Fig.2 should be smaller because of near overlapping data points.
- “In this paper” should be reduced appropriately.
- Line 175, the authors give “In the region from 0.1 to 0.3 we see an exponential dependence of the loading rate”, but for q>0.3 in Fig.3, no fit and explanation were given, why?
Author Response
Detailed response to referee 2's report
=======================================
Referee 2 states that "... the results
are interesting and can be considered for
publication in atoms."
However, the referee wants the
following 5 points to be addressed:
1. Abstract, line 2: the author should write a magneto-optic trap instead of a MOT.
2. Four dynamical regimes are not clearly defined, especially,
in Fig. 1 four dynamical regimes should be clearly marked
to increase readability.
3. Symbols in Fig.2 should be smaller because of near
overlapping data points.
4. “In this paper” should be reduced appropriately.
5. Line 175, the authors give “In the region from 0.1 to 0.3
we see an exponential dependence of the loading rate”,
but for q>0.3 in Fig.3, no fit and explanation were given, why?
These are all excellent points and I answer them
here, point-by-point, in full.
1. Answer:
----------
I agree.
Action taken:
-------------
In the abstract,
avoiding the awkward phrase "magneto-optic-trap-loaded",
I reworded the phrase
"... we investigate here the properties of the saturated,
steady state of a MOT-loaded three-dimensional Paul trap"
to:
"... we investigate here the properties of the saturated,
steady state of a three-dimensional Paul trap, loaded from
a magneto-optic trap."
This implements the referee's recommendation to replace
"MOT" by magneto-optic trap.
2. Answer:
----------
I agree.
Action taken:
-------------
The 4 dynamical regimes are now clearly marked
by Roman numerals, I, II, III, IV
in Fig. 1. The symbols added in Fig.1 are
explained in the caption of Fig. 1.
3. Answer:
----------
In fact, one of the major points
of Fig. 2 is to show that the plot symbols
overlap, corroborating the quality of the
scaling argument. However, I agree that
Fig. 2 might benefit from slightly smaller
plot symbols.
Actions taken:
--------------
I reduced the size of the plot symbols
in Fig. 2 somewhat.
The plot symbols still overlap, which
(a) cannot be avoided, lest the plot symbols
become too small to see, and
(b) the overlap is fine, and, in fact, desired,
since the overlap indicates graphically
the quality of the scaling argument.
4. Answer:
----------
I agree with the referee that the number
of occurrences of the phrase
"in this paper" needed to be reduced.
Action taken:
-------------
Following the advice of the referee,
13 occurrences of the phrase
"in this paper" were deleted.
5. Answer:
----------
This is a very interesting question.
I did not provide a fit curve, since I do
not yet have an analytical theory for the locations
of the maxima, in particular not for the region
q > 0.3.
However, qualitatively, I argue that
there is an interference between regions II
and IV, as a result of which the region-II
maxima turn into shoulders on the region-IV curve.
De-convoluting region II from region IV may
reveal the true locations of the region-II
maxima. But this has not yet been done.
Actions taken:
--------------
(a) A blue, straight line, representing the
exponential function (37) has been added
to Fig. 3 to better illustrate
the near-exponential behavior
in 0.1 < q < 0.3 and the deviation from exponential
for q > 0.3.
(b) Addressing the referee's question about
q > 0.3, making my argument above
about the possible shoulder explanation
more explicit, and also addressing the
point at q=0.05, I added the following text
at the end of subsection 2.4:
"The deviation from
exponential behavior in the region
q>0.3 may be due to the fact that
for increasing q the maximum in region II
turns into a shoulder on the region-IV curve
(see Fig. 1), masking a
possible continuation of the exponential
behavior. However, a clearer understanding
of the deviation from exponential behavior
has to await further analytical progress.
The deviation of the data point at q=0.05
has a simpler explanation. It is
due to the fact that our simulations are cut
at \lambda=0.01 and thus do not extend far enough to
the left to capture the true location of
the maximum of the loading curve at q=0.05."
Reviewer 3 Report
The manuscript is of a good quality but written in an unusual way in my opinion since usually authors first present the methods and the tools they use before their results. I found that the introduction is not "smooth" enough in the sense there is a lack of generalities and the overall context is not explained. Also the references devoted to the introduction section are limited and many concer the author and collaborators. Adding other references by other international groups may increase the quality of the manuscript. Another point of concern is that the manuscript is not self-sufficient as many parts refer to results or symbols published previously by the author and collaborators (not explained in the manuscript) as the four dynamical regions of the loading for instance. In addition, I found that a typical scheme of an experimental setup is missing even if the manuscript is of theoretical and numerical type. I found also that the simulation part is not well described especially the control parameters are not indicated and only few details are given. I think the simulations should be described more precisely.
Here are some of my detailed comments.
-- The structure should be rearranged differently to allow the presentation the the theory and the simulation techniques before presenting the results. A discussion section may be introduced as well as a conclusion one.
-- Please explain the 4 phases even if explained in your previous work and references.
-- In page 2, the author uses the term section 2 which is divided in 5 subsections but in several parts the author use the term section instead of subsection to refer to 2.1, 2.2, ...2.5 subsections.
-- Page 3, L102. I am not convinced by the the term "dangerous" .
-- Page, eq(6): U_pp is function of x only but the RHS is a function of x, y and z.
-- Line after eq (12). "charge of charge density"... --> to rephrase.
-- Eq( 13). What does Q_encl stand for ?
-- Page 5
-- Text after eq(20) and eq(21). You may use q' instead of q_e which is often used for the charge of electron like e.
-- Page 7, l141: reducing the heating rate --> reducing the ion loading rate.
-- Page 8.
-- A scheme may help making differences bteween R_cut, R_L.
-- loading-zone (uncorrect spelling)
--Page 11. R_1 and R_2 not defined. A scheme may help. Risk of confusion between R_cut, R_L, R_1 and R_2!!
--Page 12.
L183. Where do the values of R_1 and R_2 come from?
--Page 14. L273. "the confines"?. What is meant by this sentence?
Author Response
Detailed response to referee 3's report
=======================================
The referee states that the manuscript "... is of a good quality...",
but makes no explicit recommendation concerning publication
of the manuscript.
The referee has several legitimate concerns, some
of a general nature, some more specific.
The specific concerns are addressed below,
point by point.
Concerning the more general comments, the
referee points out that
(a) the manuscript is organized in an unusual way,
(b) the introduction needs improvement,
(c) references need to be added,
(d) symbols need to be explained better,
(e) a sketch of the experimental set-up is missing,
(f) the simulation part is not well described.
(a) Answer:
-----------
I agree with the referee
that the organization of the paper, especially for
a theory paper, is not optimal. However,
nothing much can be done about it since the
overall structure of the paper needs to follow
the journal style, which requires the sequence
of sections organized according to
Introduction, Results, Methods, optional Conclusions.
However, since another referee pointed out the
same problem, I created a Conclusions section
by re-grouping some of the material previously
appearing elsewhere in the paper.
(b) Answer:
-----------
I improved the introduction by adding a
new introductory sentence, pointing out the
relevance to cold chemistry, and adding
several new references referring to work
on ion-neutral collisions by several
national and international groups.
Two review papers are now also
explicitly featured.
(c) Answer:
-----------
This is a good point, also raised by another
referee. I added several new references
to the Introduction to point the reader toward
other national and international ion-neutral research groups.
A paper on neutral plasmas, pointed out by another referee
was also added and mentioned in the Conclusions section.
(d) Answer:
-----------
Without exception, all symbols occurring in the paper
are explained in the text. It is true that
improvement is always possible.
Therefore, following the suggestion of another
referee, I now also indicate the 4 dynamical
regions explicitly in Fig. 1.
(e) Answer:
-----------
This is an excellent suggestion.
Alas, since this article is part of a
special issue, and responses to
all referee reports, including the revised paper,
have to be completed and
submitted within the time frame of a week,
there is insufficient time for me
to compose such a figure.
I apologize to the referee (and future readers
of this paper) for the absence of this
potentially beneficial feature.
(f) Answer:
-----------
I checked the methods section carefully,
in particular with emphasis on the simulation part.
I found that all the information is provided
in the Methods section
that allows the reproduction of the data
displayed in Figs. 1,2, and 3.
I improved the presentation of the Methods section by
adding a paragraph that describes how the histograms
in Fig. 4 were generated.
Here is my detailed response
to individual referee comments:
Referee comment:
****************
-- The structure should be rearranged differently to allow
the presentation the the theory and the simulation techniques
before presenting the results. A discussion
section may be introduced as well as a conclusion one.
Answer:
-------
As explained above, the overall structure of the paper
needs to follow journal guidelines and cannot be changed.
A Discussion section is not needed (as suggested by
the journal guidelines), since all the necessary
discussion is already presented in the Results section
in connection with the different topics presented
and discussed there.
However, as suggested by the referee, I re-arranged some
of the material and created a new Conclusions section.
Action taken:
-------------
Material from elsewhere in the paper, especially
the lengthy introduction to the Results section
was re-organized into a Conclusions section, which
has been added to the manuscript.
Referee comment:
****************
-- Please explain the 4 phases even if explained in your previous work and references.
Answer:
-------
I agree.
Action taken:
-------------
On page 8 of the revised manuscript,
I inserted a paragraph that explains all four
dynamical regions. They are now also clearly
marked with Roman numerals, I, II, III, IV,
in Fig. 1.
Referee comment:
****************
-- In page 2, the author uses the term section 2 which is
divided in 5 subsections but in several parts the author
use the term section instead of subsection
to refer to 2.1, 2.2, ...2.5 subsections.
Answer:
-------
I agree that this is inconsistent.
However, I found only one occurrence of
"section" that should have been "subsection".
It is possible that the referee refers to
my use of the abbreviation "Sec." for
"section". I am not aware of an abbreviation for
"subsection" routinely used in physics journals.
In my experience "Sec." is used for both,
"section" and "subsection".
If this is not correct, I'm sure the
copy editor of the journal will use the
correct abbreviation.
Action taken:
-------------
I replaced the single occurrence of "section"
in the context of a subsection with
"subsection".
Referee comment:
****************
-- Page 3, L102. I am not convinced by the the term "dangerous" .
Answer:
-------
I agree.
Action taken:
-------------
This word no longer appears in the revised
version of the paper.
The corresponding sentence was slightly rewritten.
Referee comment:
****************
-- Page, eq(6): U_pp is function of x only but the RHS is a function of x, y and z.
Answer:
-------
The referee may have overlooked the arrow on top of x in the
argument of U_pp. This notation was introduced already
in equation (1). So, equation (6) is correct.
No action taken.
Referee comment:
****************
-- Line after eq (12). "charge of charge density"... --> to rephrase.
Answer:
-------
The sentence is correct and expresses the physics correctly.
However, a hyphen may clarify better what is meant.
Action taken:
-------------
I replaced "charge density" with "charge-density"
to clarify the meaning of the terms and the intent of the sentence.
Referee comment:
****************
-- Eq( 13). What does Q_encl stand for ?
Answer:
-------
Apologies. Since I'm teaching undergraduate physics,
and the symbol "Q_{encl}" ubiquitously occurs in
introductory undergraduate physics texts on
electricity, I failed to properly define this term.
Action taken:
-------------
I now introduce the sentence before
equation (13) with:
"Denoting by Q_encl the enclosed charge, ..."
Referee comment:
****************
-- Text after eq(20) and eq(21). You may use q' instead of q_e
which is often used for the charge of electron like e.
Answer:
-------
q_e is not a charge, but the Paul-trap control
parameter q in the case of electrons.
This is clear from the text.
I agree that this is slightly confusing at
first sight, but the use of q for both charge
and Paul-trap control parameter is so ingrained in the
history and current use of the symbols that
the notation cannot be changed.
Action taken:
-------------
To avoid any confusion,
I now introduce q_e and q with
the following sentence:
"This is also clear
from the fact that the value of the trap control-parameter
$q$ for electrons, $q_e$, is
related to the value of the ionic trap
control-parameter, $q$, according to"
Referee comment:
****************
-- Page 7, l141: reducing the heating rate --> reducing the ion loading rate.
Answer:
-------
I am grateful to the referee for having pointed this out.
Action taken:
-------------
heating --> loading
Referee comment:
****************
-- A scheme may help making differences bteween R_cut, R_L.
Answer:
-------
I don't understand what the referee means.
However, I think that both R_cut and R_L
are well defined in the paper.
No action taken.
Referee comment:
****************
-- loading-zone (uncorrect spelling)
Answer:
-------
I thank the referee for pointing this out.
Action taken:
-------------
I corrected the spelling.
Referee comment:
****************
--Page 11. R_1 and R_2 not defined. A scheme may help.
Risk of confusion between R_cut, R_L, R_1 and R_2!!
Answer:
-------
I don't agree with the referee.
R_1 and R_2 are clearly defined in the sentence
preceding equation (38).
I also don't see how R_cut, R_L, R_1, and R_2
could be confused, since they are clearly
different symbols.
No action taken.
Referee comment:
****************
L183. Where do the values of R_1 and R_2 come from?
Answer:
-------
As explained in the text, these values
were taken from Fig. 4, following their
definition, which is also clearly explained
in the text.
No action taken.
Referee comment:
****************
--Page 14. L273. "the confines"?. What is meant by this sentence?
Answer:
-------
The word "confines" means "borders" or "boundaries" usually
used in the sense of restricted to within some
boundaries.
I agree that this word may not easily be understood
by the reader.
Action taken:
-------------
I simplified the sentence by replacing
"... the confines of the absorbing boundary"
with "R_cut".
Reviewer 4 Report
The author theoretically studied the dynamics of loading the ions into a Paul trap from a magneto-optical trap. He verified three assumptions that are sometimes made under saturated, steady-state conditions. The presented work extends and completes the previous results published by the author. The manuscript is well written, and the presented results are relevant for ongoing experimental efforts. Therefore, I recommend the manuscript for publication in its present form with one comment to references in the instruction.
In the introduction, the author cited mostly the papers from the group of Prof. W.W. Smith as examples of the experimental investigations of ion-neutral collisions. While cited works are directly relevant to the presented results, I would suggest referring also to other experimental groups using MOT+Paul trap, e.g., Johannes Denschlag, Stefan Willitsch, Rene Gerritsma, Roi Ozeri, Eric Hudson, Takashi Mukaiyama.
Author Response
Detailed response to referee 4's report
=======================================
Referee 4 recommends the paper for publication.
The referee had only one request:
"While cited works are directly relevant to the presented
results, I would suggest referring also to other
experimental groups using MOT+Paul trap, e.g.,
Johannes Denschlag, Stefan Willitsch, Rene Gerritsma,
Roi Ozeri, Eric Hudson, Takashi Mukaiyama."
Answer:
-------
I completely agree with the referee that citing
related works from other groups both emphasizes
the relevance of the current paper and makes
it easier for the reader to find other groups
who are engaged in using hybrid MOT+Paul traps
in their ion-neutral collision experiments.
Action taken:
-------------
I now cite references from all groups mentioned
by the referee in the introduction of the paper.